

# Geriatric syndromes are potential determinants of the medication adherence status in prevalent dialysis patients

Chia-Ter Chao[1,2,3], Jenq-Wen Huang[3] and COGENT (COhort of GEriatric Nephrology in NTUH) study group

[1] Department of Medicine, National Taiwan University Hospital Jinshan Branch, New Taipei City, Taiwan
[2] Graduate Institute of Toxicology, National Taiwan University College of Medicine, Taipei, Taiwan
[3] Department of Internal Medicine, National Taiwan University Hospital, Taipei, Taiwan

## ABSTRACT

**Background.** Geriatric syndromes (GS) exhibit high prevalence in patients with end-stage renal disease (ESRD) under chronic dialysis irrespective of age. We sought to determine whether GS influences medication adherence in ESRD patients.

**Methods.** A prospective cohort of chronic dialysis patients was assembled. The presence of GS components, including frailty/prefrailty, polypharmacy, and malnutrition, were ascertained through a validated questionnaire, electronic records and chart abstraction, and laboratory tests. The severity of medication non-adherence was defined using the eight-item Morisky Medication Adherence Scale (MMAS). Multiple logistic regression analysis was performed targeting MMAS results and incorporating relevant clinical features and GS.

**Results.** The prevalence of frailty/pre-frailty, polypharmacy, and hypoalbuminemia/malnutrition among the enrolled participants was 66.7%, 94%, and 14%, respectively. The average MMAS scores in these dialysis patients were $2 \pm 1.7$ (range, 0–6), with only 15.7% exhibiting high medication adherence. Multiple regression analyses showed that the absence of frailty/pre-frailty ($P = 0.01$) were significantly associated with poorer medication adherence, while the presence of polypharmacy ($P = 0.02$) and lower serum albumin, a potential sign of malnutrition ($P = 0.03$), were associated with poor adherence in another model.

**Conclusion.** This study is among the very few reports addressing GS and medication adherence, especially in ESRD patients. Interventions targeting frailty, polypharmacy, and malnutrition might potentially improve the medication non-adherence and symptom control in these pill-burdened patients.

# INTRODUCTION

Treatment adherence implies the extent to which patients' behavior corresponds with the mutual agreement between patients and their care providers, and the scopes include but are not limited to prescribed medications, dietary advice, and diagnostic and treatment planning (*Cutler & Everett, 2010*). The assessment of adherence has been gaining increased

Corresponding author
Jenq-Wen Huang,
jenqwen@gmail.com,
007378@ntuh.gov.tw

popularity, since poor adherence to medications or treatments are linked with catastrophic health consequences and undue costs (*Conn et al.*, *2015*). Failure to take appropriate medications constitutes a major barrier to successful treatment, increases patient–physician mistrust, and predisposes patients to complications stemming from chronic illnesses (*Cutler & Everett*, *2010*; *McDonald, Garg & Haynes*, *2002*). A global estimate from the World Health Organization (WHO) suggests that medication non-adherence occurs in about 50% of patients with up to one-fourth of prescriptions never being filled due to motivation issues or patient-initiated drug holidays (*Conn et al.*, *2015*; *McDonald, Garg & Haynes*, *2002*). Moreover, the rate of non-adherence has not improved for decades, and this phenomenon has prompted the WHO to call medication non-adherence a "hidden epidemic."

In patients with end-stage renal disease (ESRD), treatment adherence is more problematic with a prevalence ranging between 22 and 74% (*Karamanidou et al.*, *2008*). Plausible reasons for this include the complex regimens prescribed, the large pill burdens, neuro-psychiatric sequels of renal failure, and the myriad of disturbing symptoms brought on by dialysis. Available studies primarily focus on adherence to diet/fluid instructions and dialysis regimens (*Sharp et al.*, *2005*; *Matteson & Russell*, *2010*); those addressing adherence to medications predominantly place their attention on phosphate-binders, the largest contributors to the daily pill count in these patients (*Karamanidou et al.*, *2008*). Indeed, non-adherence to phosphate-binders is independently associated with hyperphosphatemia, poor quality of life, and potentially higher risk of cardiovascular events among ESRD patients (*Karamanidou et al.*, *2008*; *Chiu et al.*, *2009*). Determinants and modifiers of medication non-adherence appear to be disease- and context-specific. For ESRD patients, a systematic review found that demographic factors (younger age), clinical variables (longer duration of dialysis, presence of diabetes mellitus [DM]), and psychosocial factors (personal beliefs, social support) are potential predictors of medication non-adherence (*Karamanidou et al.*, *2008*).

Geriatric syndromes (GS), multi-factorial conditions that do not readily fit into current disease entities, predispose affected individuals to increased disability, institutionalization, and higher mortality (*Inouye et al.*, *2007*). These phenotypes are not uncommon among patients with chronic kidney disease (CKD) and ESRD, since patients with renal failure are characterized by accelerated aging and a higher prevalence of age-related diseases with more rapid progression (*De Jager et al.*, *2009*; *Koopman et al.*, *2011*). We previously found that frailty, a type of GS suggesting increased vulnerability to external stressors, occurs in 32%–80% chronic dialysis patients; moreover, the presence of frailty correlates with multimorbidity and hypoalbuminemia, both critical determinants of patient survival (*Chao et al.*, *2015a*). However, whether the presence of GS plays a role in influencing medication adherence in these patients is currently unknown, since symptomatic management with medications is vital for them. We hypothesize that GS could modify the medication adherence status of chronic dialysis patients, and investigate this issue using a prospectively enrolled cohort.

## METHODS

### Study design

The current study is based on a group of prospectively enrolled elderly dialysis patients from a single center (*Chao et al.*, *2015a*; *Chao et al.*, *2015b*). The project was approved by the Institutional Review Board of National Taiwan University Hospital (No. 201403006RINB) and adheres to the Declaration of Helsinki. Participants provided verbal informed consent.

ESRD patients receiving chronic hemodialysis for more than three months were consecutively recruited from January, 2014, to September, 2015, after providing consent. The inclusion criteria were as follows: those whose age higher than 20 and received chronic dialysis for more than three months. We excluded those who were pregnant, who had dementia at baseline, or those who were not prescribed any medications in the preceding three months. On enrollment, clinical features of the participants including demographic data (age, gender, dialysis duration, educational levels, marriage status), economic condition, ESRD etiology, and baseline comorbidities were documented systematically (*Chao et al.*, *2012*; *Chao et al.*, *2015c*). All participants received blood tests for serum biochemical panels, electrolytes, nutritional and iron profiles, hemogram, and dialysis adequacy assessment after enrollment. The medication history was obtained and verified using the PharmaCloud System of the Taiwan National Health Insurance, as reported previously (*Chao et al.*, *2015d*).

### Evaluation for the presence of GS

The spectrum of GS is expanding, and the definitions of each component are constantly evolving. In the current study, we included the following GS for analysis: frailty, polypharmacy, and malnutrition, all of which have been demonstrated to influence patient outcomes in past studies (*Chao et al.*, *2015a*; *Tentori et al.*, *2007*; *Herr et al.*, *2015*). However, very few evaluated whether these geriatric phenotypes play a similarly important role among chronic dialysis patients, and none addresses the relationship between frailty and drug adherence. Frailty was assessed using the simple FRAIL scale after enrollment; this is a five-item self-report screening instrument for frailty originally introduced by the International Association of Nutrition and Aging, defined by the number of positive scores within perspectives of fatigue, resistance, ambulation, illness, and weight loss (*Morley, Malmstrom & Miller*, *2012*). Results from the simple FRAIL scale exhibits high correlation with results from other assessment tools (*Chao et al.*, *2015a*). The total scores on the simple FRAIL scale range between 0 and 5 with higher scores suggesting increasing severity. Patients were categorized as being frail if they scored higher than two out of five and as being pre-frail if they had scores of one or two. Polypharmacy was defined as the concurrent use of five or more medicines, in accordance with existing literature, while the use of 12 or more medicines qualified as extreme polypharmacy (*Herr et al.*, *2015*; *Gnjidic et al.*, *2012*). Malnutrition was defined according to serum albumin levels less than 3.5 mg/dL.

### Ascertainment of medication adherence status

In this study, we assessed these patients' adherence status after enrollment using the Taiwanese version of the eight-item Morisky Medication Adherence Scale (MMAS), a

widely adopted tool (*Morisky et al.*, *2008*; *Lin et al.*, *2013*). The original MMAS is a generic self-report scale assessing medication-taking behavior, consisting of four items: "Do you ever forget to take medications?", "Are you careless at times about taking medications?", "Sometimes, if you feel worse when you take medications, do you stop taking them?", and "When you feel better, do you sometimes stop taking your medicine?" (*Morisky, Green & Levine*, *1986*). The original four-item MMAS was recently expanded and revised into an eight-item version, which exhibits more favorable psychometric properties and clearly addresses the complex barriers to medication taking (*Morisky et al.*, *2008*). Results obtained from both the four-item and eight-item MMAS demonstrate fair correlation with those from pill counting methods. A modified Taiwanese-translated version with fair internal consistency and test–retest reliability was harnessed in this study (*Lin et al.*, *2013*). The response choices for the eight items are yes/no for the first to the seventh item and scoring on a five-point Likert scale for the last item. One point is assigned to the first seven items if the patient responds yes, or no for the fifth item, while one-fifth of a point is given for each increasing stratum within the eighth item. The total score on the eight-item MMAS ranges between 0 and 8, and higher scores indicate poorer medication adherence. Each item was verified twice by different interviewers during the patients' inter-dialytic period.

## Statistical analysis

We performed statistical analyses using SPSS 18.0 software (SPSS Inc., Chicago, IL, USA). Continuous variables were expressed as means $\pm$ standard deviations and compared between groups with an independent $t$-test or a Mann–Whitney $U$-test (if not satisfying normal distribution). Categorical variables were shown as event counts (percentages in parentheses) and compared between groups using Chi-square tests. We first assessed the correlation between MMAS results (as linear variables) and the clinical/laboratory features among the entire cohort using Pearson's correlation coefficients. Finally, we conducted multivariate linear regression analyses with the MMAS results as the dependent variable, incorporating demographic variables (age and gender), DM or not, serum albumin levels, as well as the components of GS. In all analyses, two-sided $P$ values less than 0.05 were considered statistically significant.

## RESULTS

### Clinical features of the cohort

Among 52 patients who were screened, only one was excluded due to the absence of chronic medication use. A total of 51 patients were then prospectively recruited. The mean age of the enrolled chronic dialysis patients was 68 $\pm$ 11.8 years with nearly half being men (Table 1). About 40% of enrollees had received elementary school education only, and most participants were married. The average dialysis duration was 3.4 $\pm$ 2.8 years. 14 (27.5%) patients needed service from caregivers.

More than half of these patients had DM at enrollment, and most were hypertensive (Table 1). The mean dialysis comorbidity index of this cohort, using Liu et al.'s method, was 1.6 $\pm$ 1.8, similar to the previous reports on Taiwanese ESRD patients (*Chao et al.*, *2015a*; *Chao et al.*, *2012*). About 40% of the ESRD resulted from diabetic nephropathy.

**Table 1** **Clinical features of the enrolled chronic dialysis patients.** Data are expressed as mean ± standard deviation for continuous variables, and number (percentage) for categorical variables.

| Clinical features | Total |
|---|---|
| *Demographic profiles* | |
| Age (years) | 68 ± 11.8 |
| Gender (male %) | 22 (43) |
| *Education* | |
| None | 19 (37) |
| Elementary school | 21 (41) |
| Junior or senior high school | 7 (14) |
| College or higher | 4 (8) |
| Marriage status (yes %) | 47 (92) |
| Monthly disposable incomes (higher than 300 USD) | 24 (47) |
| Duration of dialysis (years) | 3.4 ± 2.8 |
| *Comorbidities (%)* | |
| Diabetes mellitus | 26 (51) |
| Hypertension | 46 (90) |
| Cirrhosis | 11 (22) |
| Heart failure | 4 (8) |
| Malignancy | 6 (12) |
| *Dialysis comorbidity index* | 1.6 ± 1.8 |
| *Etiology of ESRD (%)* | |
| Diabetic nephropathy | 21 (41) |
| Chronic glomerulonephritis | 4 (8) |
| Miscellaneous | 7 (14) |
| Unknown | 19 (37) |
| *Laboratory parameters* | |
| *Biochemistry and electrolytes* | |
| Blood Urea Nitrogen (mg/dL) | 80.8 ± 19.9 |
| Creatinine (mg/dL) | 10.6 ± 2.5 |
| Sodium (meq/L) | 135 ± 3.9 |
| Potassium (meq/L) | 4.7 ± 0.7 |
| Calcium (mg/dL) | 8.9 ± 0.8 |
| Phosphate (mg/dL) | 5.3 ± 1.5 |
| Calcium-phosphate product | 47.3 ± 14 |
| *Nutrition* | |
| Albumin (mg/dL) | 3.8 ± 0.3 |
| Glucose (mg/dL) | 107 ± 36 |
| Total cholesterol (mg/dL) | 158 ± 40 |
| Total triglyceride (mg/dL) | 157 ± 108 |
| *Anemic profile* | |
| Hemoglobin (g/dL) | 9.7 ± 1.3 |
| Ferritin (ng/mL) | 647 ± 784 |
| Transferrin saturation (%) | 28.3 ± 15 |

**Table 1** (*continued*)

| Clinical features | Total |
|---|---|
| *Dialysis adequacy* | |
| Single pool $Kt/V$ | $1.6 \pm 0.2$ |
| Urea reduction ratio | $74.7 \pm 5.9$ |

**Notes.**

Abbreviations: ESRD, End-stage renal disease; USD, United States Dollar.

**Table 2** List of medications used.

| Medication class (prevalence %) | Total |
|---|---|
| $\alpha$ - blockers | 12 (24) |
| $\beta$ - blockers | 14 (27) |
| ACEi/ARBs | 15 (29) |
| Diuretics | 11 (22) |
| BZDs | 28 (55) |
| Anti-psychotics | 3 (6) |
| Anti-depressants | 8 (16) |
| Phosphate-binders | 43 (84) |
| Potassium-binding resins | 13 (25) |
| *Total medication count (pills per day)* | $12.1 \pm 5.2$ |
| *Total anti-hypertensive medication count (pills per day)* | $1.5 \pm 1.9$ |
| **Polypharmacy** | 48 (94) |
| **Extreme Polypharmacy** | 27 (53) |

**Notes.**

Abbreviations: ACEi, Angiotensin-converting enzyme inhibitor; ARB, Angiotensin receptor blocker; BZD, Benzodiazepine.

On average, these chronic dialysis patients were prescribed $12.1 \pm 5.2$ pills daily and took $1.5 \pm 1.9$ anti-hypertensive medications (Table 2). One-fourth of these patients were taking $\alpha$- and $\beta$-blockers while more than half were using benzodiazepines. Anti-psychotics and anti-depressants were both given in less than 20% of patients.

## The prevalence of GS

All the patients responded to simple FRAIL questionnaire by themselves. Using the simple FRAIL scale, we found that the average frailty severity score of these patients was $1.4 \pm 1.3$ (maximal score, 5). 10 (19.6%) patients were categorized as frail, and 24 (47.1%) were pre-frail, resulting in two-thirds of participants being either frail or pre-frail. Polypharmacy occurred in 48 (94%) patients, and extreme polypharmacy was found in 27 (53%) patients. Serum laboratory profiles are shown in Table 1. The average serum albumin and hemoglobin levels were $3.8 \pm 0.3$ mg/dL and $9.7 \pm 1.3$ g/dL, respectively. Serum albumin levels were higher than 3.5 mg/dL in 44 (86%) patients.

## Medication adherence status

All the patients responded to the MMAS questionnaires by themselves. The mean MMAS score was $2 \pm 1.7$ with a range between 0 and 6. Only 15.7% of patients manifested high adherence (defined as MMAS score 0). On average, 39%, 14%, 45%, 25%, 33%, and 10%

of patients responded "yes" to the first ("sometimes forget to take medicines"), second ("ever forgot in recent two weeks"), third ("self-adjust regimens if uncomfortable"), fourth ("forget to bring medications during travel"), sixth ("self-adjust regimens in stable conditions"), and seventh items ("feel disturbed for trying to remember to take medicines"), respectively. On the other hand, 10% of patients responded "no" to the fifth item ("did you remember to take medicines yesterday").

## Correlation between MMAS results and clinical/laboratory parameters and the comparison between patients with high and low medication adherence

Significant correlation was found between MMAS scores and several clinical variables we collected. We found that MMAS scores were negatively associated with age ($r = -0.3$, $P = 0.03$), simple FRAIL scale scores ($r = -0.28$, $P = 0.045$), and $Kt/V$ ($r = -0.29$, $P = 0.04$). Furthermore, male patients had significantly higher MMAS scores than did female patients (the former vs. the latter, $2.5 \pm 2.0$ vs. $1.6 \pm 1.4$, $P = 0.04$). Frail and pre-frail patients had lower MMAS scores than did non-frail patients (the former vs. the latter, $1.6 \pm 1.5$ vs. $2.9 \pm 1.9$, $P = 0.02$). Patients with extreme polypharmacy had a trend of higher MMAS scores compared to those without (the former vs. the latter, $2.18 \pm 1.85$ vs. $1.79 \pm 1.61$, $P = 0.44$).

A comparison of clinical features between chronic dialysis patients with high and low medication adherence is provided in Table 3. No significant differences were observed among those with high and low medication adherence status in the tested variables, except calcium-phosphate products ($P = 0.04$).

## Regression analyses for the determinants of medication adherence status

Finally, we performed multivariate linear regression analyses to identify the potentially independent factors influencing medication non-adherence severity in ESRD patients under chronic dialysis (Table 4). With MMAS score as the dependent variable, we discovered that the presence of frailty/pre-frailty was independently associated with lower MMAS scores (better adherence) ($P = 0.01$). We further analyzed the relationship incorporating polypharmacy or not, and found that the presence of polypharmacy was independently associated with higher MMAS score (poor adherence) ($P = 0.02$), while serum albumin levels presented an inverse linear association with the MMAS scores ($P = 0.03$). However, extreme polypharmacy did not exhibit significant relationship with the MMAS scores ($P = 0.17$). We also conducted a logistic regression analysis with high or low medication adherence status as the dependent variable, but all the variables showed insignificant associations with the medication adherence status. This might suggest that treating the MMAS score as a continuous variable is more suitable for investigating the determinants of the medication adherence status in these patients. Finally, sensitivity analyses including only patients with DM yielded essentially similar findings to those using the linear regression analyses.

**Table 3** Comparison of chronic dialysis patients with high and low medication adherence status.

| Clinical features | High adherence (n = 8) | Low adherence (n = 43) | p value |
|---|---|---|---|
| *Demographic profiles* | | | |
| Age (years) | 71.1 ± 9.3 | 67.4 ± 12.2 | 0.42 |
| Gender (male %) | 4 (50) | 18 (42) | 0.68 |
| Duration of dialysis (years) | 4.5 ± 4.1 | 3.2 ± 2.5 | 0.23 |
| Monthly disposable income (higher than 300 USD) | 5 (63) | 19 (44) | 0.35 |
| *Comorbidities (%)* | | | |
| Diabetes mellitus | 4 (50) | 22 (51) | 0.95 |
| Hypertension | 7 (88) | 39 (91) | 0.79 |
| Cirrhosis | 1 (13) | 3 (7) | 0.6 |
| Heart failure | 2 (25) | 9 (21) | 0.8 |
| Malignancy | 2 (25) | 4 (9) | 0.21 |
| *Dialysis comorbidity index* | 2 ± 1.9 | 1.6 ± 1.8 | 0.52 |
| *Etiology of ESRD (%)* | | | 0.81 |
| Diabetic nephropathy | 3 (38) | 18 (42) | |
| Chronic glomerulonephritis | 1 (13) | 3 (7) | |
| Miscellaneous | 0 (0) | 7 (16) | |
| Unknown | 4 (50) | 15 (35) | |
| *Laboratory parameters* | | | |
| *Biochemistry and electrolytes* | | | |
| Blood Urea Nitrogen (mg/dL) | 73.5 ± 13.2 | 82.2 ± 20.8 | 0.26 |
| Creatinine (mg/dL) | 10.7 ± 2.2 | 10.6 ± 2.5 | 0.99 |
| Sodium (meq/L) | 136 ± 4.1 | 135 ± 3.9 | 0.61 |
| Potassium (meq/L) | 4.8 ± 0.8 | 4.7 ± 0.7 | 0.65 |
| Calcium (mg/dL) | 8.6 ± 0.9 | 9 ± 0.8 | 0.25 |
| Phosphate (mg/dL) | 4.3 ± 1.3 | 5.5 ± 1.5 | 0.06 |
| Calcium-phosphate product | 37.9 ± 3.7 | 49.1 ± 13.5 | 0.04 |
| *Nutrition* | | | |
| Albumin (mg/dL) | 3.8 ± 0.4 | 3.8 ± 0.3 | 0.97 |
| Glucose (mg/dL) | 87.5 ± 19.9 | 110.6 ± 39.1 | 0.11 |
| Total cholesterol (mg/dL) | 154 ± 41 | 158 ± 40 | 0.77 |
| Total triglyceride (mg/dL) | 119 ± 87 | 165 ± 111 | 0.28 |
| *Anemic profile* | | | |
| Hemoglobin (g/dL) | 9.3 ± 0.4 | 9.8 ± 1.4 | 0.35 |
| Ferritin (ng/mL) | 529 ± 172 | 670 ± 852 | 0.65 |
| Transferrin saturation (%) | 31.7 ± 16.6 | 27.6 ± 14.8 | 0.48 |
| *Dialysis adequacy* | | | |
| Single pool $Kt/V$ | 1.6 ± 0.2 | 1.6 ± 0.3 | 0.99 |
| Urea reduction ratio (%) | 75 ± 3.5 | 75 ± 6.3 | 0.9 |

**Notes.**
Abbreviations: ESRD, End-stage renal disease; USD, United States Dollars.

**Table 4  Results from linear regression analyses, with the Morisky's medication adherence scale score as the dependent variable.**

| Results | t value | β coefficient | p value |
|---|---|---|---|
| *Model 1* | | | |
|     Presence of frailty and pre-frailty | −2.87 | −0.37 | 0.01 |
| *Model 2* | | | |
|     Presence of polypharmacy | 2.41 | 0.32 | 0.02 |
|     Presence of frailty and pre-frailty | −3.84 | −0.51 | <0.01 |
|     Serum albumin (per mg/dL) | −2.27 | −0.32 | 0.03 |

**Notes.**

Model 1 included variables from age, gender, DM or not, serum albumin level, and the presence of frailty/pre-frailty or not.
Model 2 included variables from Model 1 and the polypharmacy.
Abbreviations: DM, Diabetes mellitus.

## DISCUSSION

In the current study, we found that more than 80% of chronic dialysis patients had low medication adherence, and the severity of non-adherence increased significantly with younger age, lower degree of frailty, and lower dialysis clearance. Furthermore, the components of GS, including frailty, malnutrition, and polypharmacy, were significant modifiers of medication adherence status in these patients (Fig. 1). These findings shed light on the potential relationship between GS and medication adherence in the ever-growing ESRD population with high pill burden.

The rate of non-adherence in the current cohort is similar to that found in existing reports. Through monitoring serum phosphate levels or self-rating scales for assessment, a systematic review concluded that the rate of non-adherence to phosphate binders falls between 46% and 74% (*Karamanidou et al.*, *2008*). Recently, Chiu and colleagues identified 62% of dialysis patients as non-adherent to their medications, half of which consisted of phosphate-binders (*Chiu et al.*, *2009*). In addition to phosphate-lowering medications, the non-adherence rate for anti-hypertensive drugs was also higher than 50% in chronic dialysis patients (*Neri et al.*, *2011*). Another cohort study similarly revealed that only 40% of ESRD patients were perfectly adherent to their prescribed medications (*Rosenthal Asher, Ver Halen & Cukor*, *2012*). Furthermore, in the same study, non-adherence was significantly predictive of higher mortality over five years of follow-up (*Rosenthal Asher, Ver Halen & Cukor*, *2012*). Consequently, this high prevalence of medication non-adherence, combined with its potential prognostic importance in ESRD patients, calls for further research into the factors associated with non-adherence.

A novel finding of this study is that the degree of frailty is inversely associated with that of medication non-adherence (Fig. 1). Frailty denotes a state of increased vulnerability to external stressors, presumably resulting from a decline of age-related physiologic reserve. In chronic dialysis patients, frailty has been found to be significantly correlated with hypoalbuminemia and inflammatory status, both of which are important modifiers of patient outcomes; presence of frailty also increases the adverse outcomes in these patients (*Chao et al.*, *2015a*; *Alfaadhel et al.*, *2015*). Studies addressing the relationship between

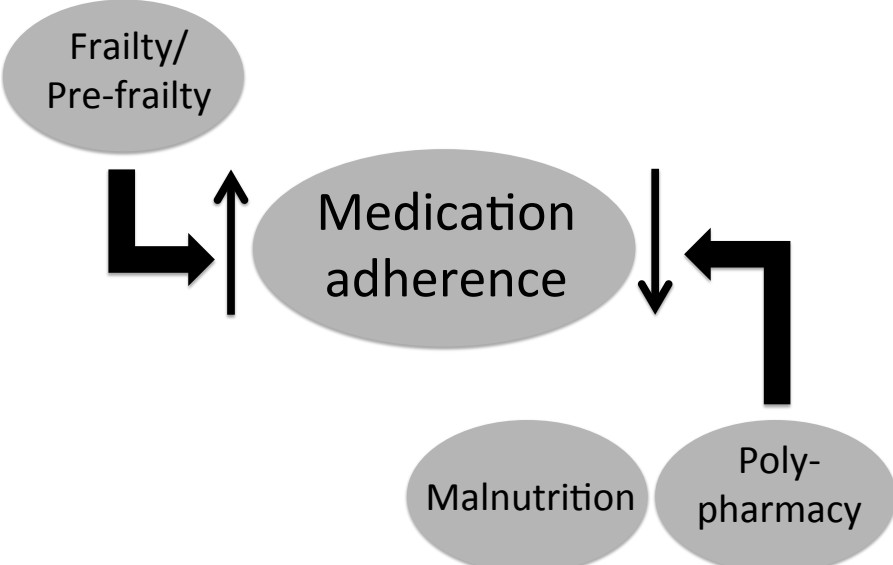

**Figure 1 Schematic diagram illustrating the identified relationship between medication adherence and components of the geriatric syndrome among end-stage renal disease patients under chronic dialysis.**

adherence and frailty predominantly involve dietary or therapy adherence rather than medication adherence, and the target population is heterogeneous. Some reports suggest that frailty is variably associated with lower dietary adherence, while others identify an absence of association (*Corsonello et al.*, *2009*; *Talegawkar et al.*, *2012*). Our finding in dialysis patients has not been reported previously and further elucidates an under-recognized influence brought by frailty in ESRD patients. It might seem counter-intuitive that frail dialysis patients have better adherence to their medications, but several plausible reasons exist to explain this association. First, frail dialysis patients tend to be older, and increasing age is an established predictor of better medication adherence (*Karamanidou et al.*, *2008*). Elderly individuals may be more concerned about their survival, uphold the necessity of adherence, and be more willing to accommodate the prescribed schedules than are their younger counterparts (*Boyer et al.*, *1990*; *Horne et al.*, *2013*). On the other hand, another possibility is that frail patients might be too weak or cognitively impaired to report their adherence status as accurately as their younger counterparts (*Wu et al.*, *2015*). In our analysis, this association stands after adjusting for age and other sociodemographic variables (Table 4), lending support to the validity of this finding. Further investigation is needed to elucidate the potential mechanisms linking frailty and medication adherence status in ESRD patients under chronic dialysis.

A high pill burden is expectedly an important barrier to adherence, as a systematic review revealed that the prescribed number of daily doses and dosing frequencies are inversely associated with treatment and medication adherence (*Claxton, Cramer & Pierce*, *2001*). Hypoalbuminemia, a potential token of malnutrition, is associated with lower medication adherence in chronic dialysis patients (Table 4). Dialysis patients with protein-energy

malnutrition tend to exhibit physical unfitness and impaired ability to comply with their medication schedules. For other diseases, malnutrition is not only a significant outcome predictor but also an important modifier of medication adherence (*Berhe, Tegabu & Alemayehu*, *2013*); in chronic dialysis patients, non-compliance frequently co-exists with hypoalbuminemia and lower body weight (*Ibrahim, Hossam & Belal*, *2015*). The association between serum albumin and the severity of medication non-adherence might also be explained by the occult musculoskeletal influence introduced by hypoalbuminemia.

There are multiple types of geriatric syndrome identified to date, but we only studied frailty, polypharmacy, and malnutrition in this report. Our research group previously spent much effort evaluating the influence of frailty and polypharmacy in patients under chronic dialysis or geriatric patients (*Chao et al.*, *2015a*; *Chao et al.*, *2015d*; *Chao, Huang & Chiang*, *2016*). Since the validation of tools used to ascertain frailty, polypharmacy, and hypoalbuminemia/malnutrition has been done in the current cohort before, we choose to include these three geriatric components in our analysis. On the other hand, for chronic dialysis patients, urinary incontinence cannot be determined, since most of them are anuric. The frequency of fall among these chronic dialysis patients is also very low, precluding further analysis. We are currently in the process of assessing other components of geriatric syndromes, such as delirium, and the results would be available in the future.

The current study has its strengths and limitations. The focus of our study addresses an important but under-reported issue, the presence of GS and medication adherence status. We affirmed the independent relationship between each GS component and self-report non-adherence to medications. Despite this advantage, our cohort is still limited by the modest case number and the unclear generalizability of our findings to population of different ethnic origins or that with different illnesses. Furthermore, performance-based measurement of frailty was unavailable among these patients. However, as we have comprehensively adjusted for interfering variables, we believe that our findings are important and warrant further studies to uncover the related mechanisms.

## CONCLUSION

Renal failure is often regarded as an accelerated aging process. GS, phenotypes that characterize different health statuses in elderly individuals, are important geriatric outcome determinants and might play a similar role in chronic dialysis patients. We found that less severe frailty, polypharmacy, and malnutrition, all components of GS, are significantly associated with medication non-adherence in these patients. Interventions targeting these phenotypes, especially actions led by clinical pharmacists against polypharmacy, medication reconciliation, and nutritional enhancement might be beneficial for chronic dialysis patients to improve their medication adherence and achieve better symptomatic control.

### List of abbreviations

| | |
|---|---|
| **CKD** | Chronic kidney disease |
| **DM** | Diabetes mellitus |
| **ESRD** | End-stage renal disease |

| GS | Geriatric syndrome |
|---|---|
| **MMAS** | Morisky Medication Adherence Scale |
| **WHO** | World Health Organization |

## ACKNOWLEDGEMENTS

We are grateful to the assistance in data collection from all staffs and nurses in the National Taiwan University Hospital Jin-Shan Branch.

### Funding

This study is financially supported by a National Taiwan University Hospital grant (No. 105-N3206). The funders had no role in study design, data collection and analysis, decision to publish, or preparation of the manuscript.

### Grant Disclosures

The following grant information was disclosed by the authors:
National Taiwan University Hospital: 105-N3206.

### Competing Interests

The authors declare there are no competing interests.

### Author Contributions

- Chia-Ter Chao conceived and designed the experiments, performed the experiments, analyzed the data, contributed reagents/materials/analysis tools, wrote the paper, prepared figures and/or tables, reviewed drafts of the paper.
- Jenq-Wen Huang performed the experiments, analyzed the data, contributed reagents/materials/analysis tools, wrote the paper, reviewed drafts of the paper.

### Human Ethics

The following information was supplied relating to ethical approvals (i.e., approving body and any reference numbers):

The project was approved by the Institutional Review Board of National Taiwan University Hospital (No. 201403006RINB) and adheres to the Declaration of Helsinki.

### Data Availability

The raw data has been supplied File S1.

### Supplemental Information

Supplemental information for this article can be found online at http://dx.doi.org/10.7717/peerj.2122#supplemental-information.

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
