# Peer review of "Geriatric syndromes are potential determinants of the medication adherence status in prevalent dialysis patients"

_PeerJ, doi:10.7717/peerj.2122_

## Round 0.1 · original submission · Minor Revisions

Please, note that we had difficulties in finding reviewers for your submission (many were contacted but most of them refused or did not reply to our invitation). Yet, we eventually got a detailed and constructive review from an expert in the field.

I also reviewed the paper myself and found it interesting but in need of improvement along the lines suggested by the reviewer. We, therefore, expect you to take these remarks and suggestions in full consideration and to come with a detailed (point by point) rebuttal when submitting a revised version.

·

Basic reporting

One minor comment:
The figure explaining the relationships between medication adherence and frailty/prefrailty on one hand and malnutrition or polypharmacy on the other, might suggest a possible generalisation of the model. Instead, the question of the generalizability shoud be addressed in the limitations section of the discussion. The legend should include the population addressed in the study (ESRD patients on dialysis)

Experimental design

The experimental design is well described. The main tools used (FRAIL, MMAS) are simple and efficient assessments, as self administred questionnaires there is a possible subjective or cognitively influence. Are there objective data available to strengthen these scores (Edmonton?). Regarding the definition of polypharmacy >5 drugs, 94% are concerned. Were the data analyzed for extreme polypharmacy (52% of the sample)?
Regarding the interpretation of the concept of geriatric syndromes, the authors included frailty in the GS. Cognitive impairment is a shared risk factor for frailty (Inouye 2007) and since the data collected do not include assessment of cognitive status, does it mean that dementia was an exclusion criteria? The mention "no patient had dementia" need clarification in that respect.

Validity of the findings

The geriatric syndromes choosed to explain drug adherence are limited to frailty, polypharmacy and malnutrition. There are important incidental other geriatric syndromes not addressed such as falls (frequent in dialysis elderly), delirium or incontinence. The reason these geriatric indicators have not been considered needs a comment in the discussion. The Edmonton frail scale the authors used in previous studies is not used here, the authors use their own validated simple FRAIL-K scale which to our knowledge has not been validated to outcomes of frailty (see also the above comment on the self administred questionnaires).

Additional comments

This is a well conducted study on ESRD patients on dialysis by the team of the authors who previously made several other researches on frailty in these patients particularity the older ones. The study addresses the medication adherence which is a major challenge in these patients taking an average of 12 pills, which is much more than in older medicalized cohorts (average 5 to 8 different drugs). The hypothesis that geriatric syndromes and frailty status could modulate drug adherence is interesting in this population. The fact that the subjects of this study are selected from a younger age than the classical geriatric population (>20 years of age, mean age 68±11 ; >75 for a geriatric sample) could explain the main paradox in the analyses of the results : more frail have better adherence and not frail are less adherent. It looks like there are two distinct populations in the sample: those getting better standard treatment and drug council?, and those not frail with other health needs and more self-assisted?. Nevertheless, it is difficult to interpret this interesting contra-intuitive observation, due to the size and the possible heterogeneity of the population. Therefore, if the originality of this study is certain (effect of geriatric syndromes on drug adherence in ESRD dialyzed patients), the conclusion should be clear about the paradox observed, and the second sentence of the conclusion schould be better phrased as « We found that less severe frailty, polypharmacy, and malnutritiond... ». Regarding the possible interventions to improve adherence, polypharmacy and malnutrition are effective targets, however frailty looks more like an indicator to look at the risks of polypharmacy in the less frail. To address this question of different health needs across this population, the intervention of a clinical pharmacist would be recommended to make a more appropriate prescription.

---

## Round 0.2 · accepted · Accept

You have reasonably taken the comments of the reviewer in consideration and your paper appears stronger as presented now. I have reviewed it, and I am happy to accept it.